# Structural basis for the phase separation of the chromosome passenger complex

Nikaela W Bryan[1,2], Aamir Ali[3], Ewa Niedzialkowska[3], Leland Mayne[1], P Todd Stukenberg[3], Ben E Black[1,2,4,5]*

[1]Department of Biochemistry and Biophysics, Perelman School of Medicine, University of Pennsylvania, Philadelphia, United States; [2]Graduate Program in Biochemistry and Molecular Biophysics, University of Pennsylvania, Philadelphia, United States; [3]Department of Biochemistry and Molecular Genetics, University of Virginia, Charlottesville, United States; [4]Penn Center for Genome Integrity, Perelman School of Medicine, University of Pennsylvania, Philadelphia, United States; [5]Epigenetics Institute, Perelman School of Medicine, University of Pennsylvania, Philadelphia, United States

*For correspondence: blackbe@pennmedicine.upenn.edu

Competing interest: The authors declare that no competing interests exist.

**Abstract** The physical basis of phase separation is thought to consist of the same types of bonds that specify conventional macromolecular interactions yet is unsatisfyingly often referred to as 'fuzzy'. Gaining clarity on the biogenesis of membraneless cellular compartments is one of the most demanding challenges in biology. Here, we focus on the chromosome passenger complex (CPC), that forms a chromatin body that regulates chromosome segregation in mitosis. Within the three regulatory subunits of the CPC implicated in phase separation – a heterotrimer of INCENP, Survivin, and Borealin – we identify the contact regions formed upon droplet formation using hydrogen/deuterium exchange mass spectrometry (HXMS). These contact regions correspond to some of the interfaces seen between individual heterotrimers within the crystal lattice they form. A major contribution comes from specific electrostatic interactions that can be broken and reversed through initial and compensatory mutagenesis, respectively. Our findings reveal structural insight for interactions driving liquid-liquid demixing of the CPC. Moreover, we establish HXMS as an approach to define the structural basis for phase separation.

## Editor's evaluation

This study is important for the phase separation field as it demonstrates that hydrogen/deuterium-exchange mass spectrometry (HXMS) can identify key regions important in driving liquid-liquid demixing. The authors convincingly confirm their HXMS results by mutagenesis. The study uses the chromosomal passenger complex (CPC) as an example, but the methodology will be applicable to other proteins or protein complexes undergoing liquid-liquid demixing.

## Introduction

Membraneless intracellular compartmentalization is central to a long and growing list of biochemical transactions at diverse sub-cellular locations (*Banani et al., 2017*). Proteins are by-and-large the drivers of the formation of these compartments, but there is debate about whether their underpinnings are either low-affinity/low-specificity interactions yielding phase separation (*Brangwynne, 2013*) or multivalent site-specific interactions (*Musacchio, 2022*). By its very nature, ascertaining the former type of interaction is essentially intractable by conventional structural biology methodologies. Moreover, the latter type of interaction is typically beyond current structural approaches, since

multivalency involving contacts with highly flexible surfaces confounds traditional methodologies used in structural studies. Approaches that provide mechanistic insight on specific complexes engaged in membraneless compartmentalization are highly limited to date, especially when the molecules undergoing putative phase separation are more complex than a relatively small individual polypeptide. There is some reported success with crosslinking approaches (*Kato et al., 2012*) and NMR (*Conicella et al., 2020*; *Conicella et al., 2016*; *Gibbs et al., 2020*) but deciphering the physical basis of membraneless compartmentalization will ultimately require advancing new technologies and/or new applications of existing ones. The proposed types of bonds that are involved in phase separation are similar to those involved in conventional protein folding and interactions (e.g. hydrophobic and electrostatic interactions) (*Banani et al., 2017*); however, specifically how the structure and dynamics of a protein or protein complexes are impacted upon engaging in higher-order interactions is almost entirely unknown.

One proposed cellular compartment is the inner centromere, comprised, in part, of chromatin and the chromosome passenger complex (CPC) (*Trivedi et al., 2019*). The CPC is one of the key regulators of cell division and is comprised of four subunits: the serine/threonine enzymatic core Aurora B kinase, and three regulatory and targeting subunits, the scaffold inner centromere protein (INCENP), Survivin, and Borealin (also known as Dasra-B) (*Ruchaud et al., 2007*). The activity of the CPC is strongly based on its sub-cellular localization during specific stages of cell division. In particular, during prometaphase the CPC is strongly localized to the chromatin spanning the two replicated centromeres, called the inner centromere. At the centromere, the CPC is involved in the process of mitotic error correction, whereby misattachments of centromeres to the microtubule-based mitotic spindle are rectified (*Lampson and Grishchuk, 2017*). Conventional targeting mechanisms through molecular recognition are required for CPC localization to the inner centromere. Specific chromatin marks at the inner centromere are recognized to direct CPC localization: the Survivin subunit directly binds H3$^{T3phos}$ and the adaptor protein, Sgo2, indirectly binds to H2A$^{T120phos}$ (*Kelly et al., 2010*; *Wang et al., 2010*; *Yamagishi et al., 2010*). The three non-catalytic subunits of the CPC (INCENP$^{1-58}$, Borealin, and Survivin) form soluble heterotrimers that have a propensity to undergo liquid-liquid phase separation (*Trivedi et al., 2019*). Deletion of one region of Borealin between amino acids 139–160 (Borealin$^{\Delta139-160}$) or disrupting the strong positive charge in this region disrupts phase separation in vitro. These mutations within Borealin also reduce CPC accumulation at the inner centromere and its ability to robustly bundle spindle microtubules (*Niedzialkowska et al., 2024*; *Trivedi et al., 2019*). Furthermore, this region of Borealin overlaps with its mapped protein surface that contributes to nucleosome binding of the CPC (*Abad et al., 2019*). Besides the requirement for this region of Borealin, nothing mechanistic is known regarding how the CPC phase separates.

Here, we measure the change in polypeptide backbone dynamics of the INCENP/ Borealin/ Survivin heterotrimer (ISB) in either a soluble or liquid-liquid demixed state using hydrogen/deuterium exchange coupled to mass spectrometry (HXMS). The most prominent changes in backbone dynamics are measured as additional protection from HX, primarily localized to discrete portions of α-helices of the INCENP and Borealin subunits. By combining information learned from peptide mapping provided by the HXMS data, a stepwise candidate mutagenesis approach, high-resolution structural information from the crystal packing behavior of ISB heterotrimers, and biochemical complementation, we identify three separate salt-bridges that drive liquid-liquid demixing.

## Results

### HXMS identifies regions with ISB heterotrimers impacted by phase separation

The ISB heterotrimer is comprised of the N-terminal 58 amino acids of INCENP, along with both full-length Survivin and Borealin (*Figure 1A*). Together, it forms a three-helix bundle, containing a histone-binding module from the Survivin subunit and a C-terminal extension of Borealin that is reported to be mostly unstructured (*Jeyaprakash et al., 2007*). Prior ISB phase separation was performed by either the addition of a polymeric crowding agent or by lowering the ionic strength (*Trivedi et al., 2019*). Polymers, like those typically used in phase separation studies (e.g. polyethylene glycol or dextran), are incompatible with the mass spectrometry step in HXMS that we intended to use to study the ISB, since the resulting spectra from polymers obscure those from the peptides under investigation. Thus,

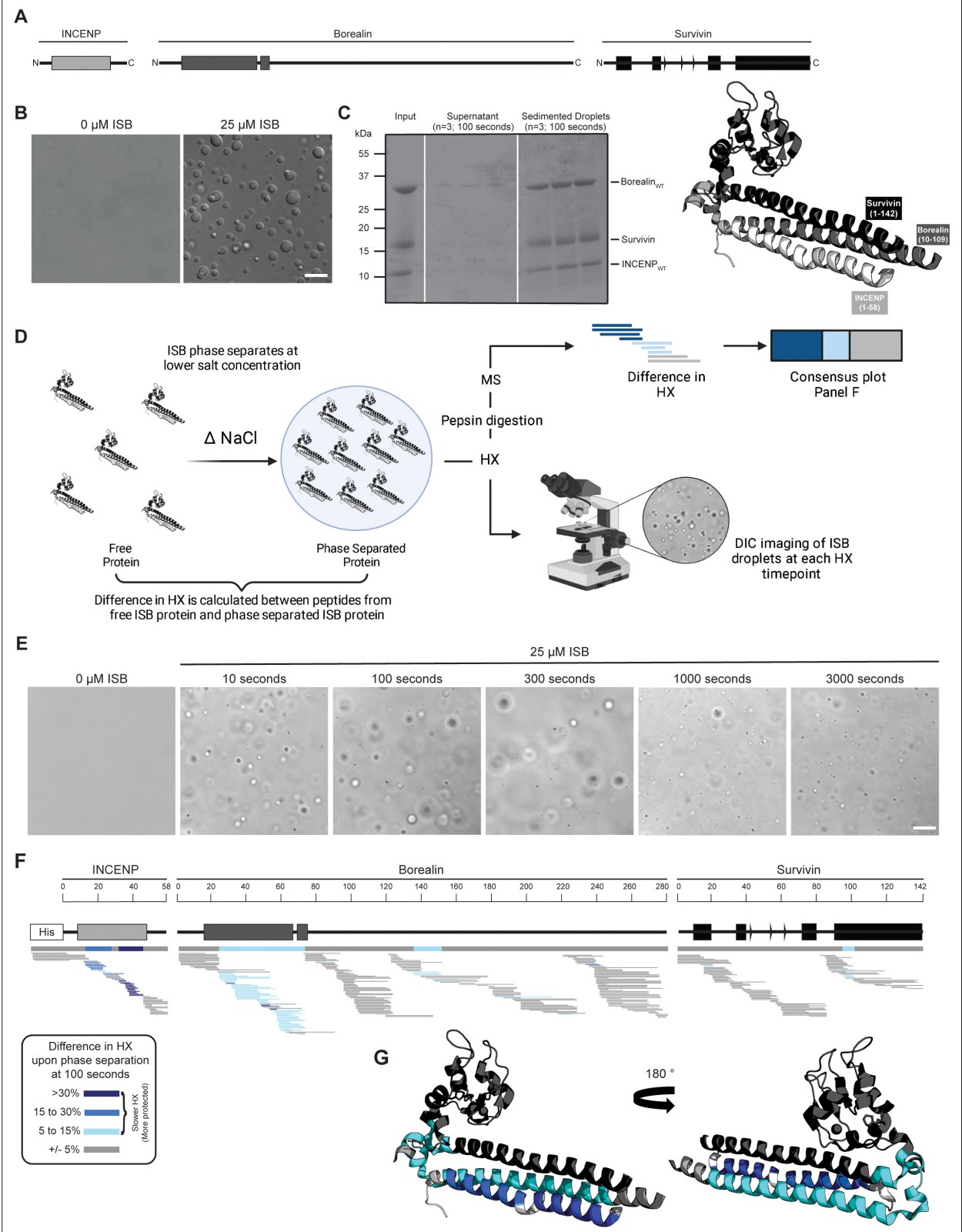

**Figure 1.** Phase separation leads to decreases in hydrogen/deuterium exchange (HX) within the three-helix bundle structure of ISB. (**A**) Schematic of the ISB showing various structural domains within the subunits. Structural information was extracted from crystal structure of three-helix bundle structure of the ISB (PDB# 2QFA) (**Jeyaprakash et al., 2007**). Each protein is color-coded in a various shade of gray: INCENP = light gray, Borealin = mid-gray, Survivin = dark gray. (**B**) DIC micrographs of the ISB droplets under indicated conditions (25 μM ISB, 75 mM NaCl). Droplets were allowed to settle onto

*Figure 1 continued on next page*

*Figure 1 continued*

coverslip before imaging (~5 min). Scale bar = 10 µm. (**C**) Sedimentation of pre-formed ISB droplets at phase separation conditions in Panel B (n=3). The time shown indicates the incubation period prior to sedimentation. (**D**) Schematic of hydrogen/deuterium exchange mass spectrometry (HXMS) experiments between free ISB protein and droplet ISB protein. HX samples either underwent pepsin digestion and analysis by MS or DIC imaging at each HX timepoint. (**E**) DIC micrographs of the ISB droplets at each HX timepoint (10, 100, 300, 1000, and 3000 s). Droplets were not allowed to settle onto coverslip to allow for accurate timing of images. Scale bar = 10 µm. (**F**) Percent difference in HX is calculated for each peptide (represented by horizontal bars) at the 100 s timepoint and plotted using the corresponding color key. The consensus behavior at each ISB residue is displayed in the horizontal bar below the secondary structure annotation taken from Panel A. These peptides were identified in a single experiment. When available, we present the data for all measurable charge states of the unique peptides within the experiment. (**G**) Consensus HXMS data from Panel F is mapped onto the three-helix bundle structure of the ISB, along with corresponding color key. Two views are shown, rotated by 180°.

The online version of this article includes the following source data, source code, and figure supplement(s) for figure 1:

**Source code 1.** MATLAB script to produce difference plots between two hydrogen/deuterium exchange mass spectrometry (HXMS) datasets.

**Source data 1.** Data used to generate *Figure 1*.

**Figure supplement 1.** Phase properties of WT-ISB in absence of crowder.

we studied the phase separation properties of the ISB over a range of protein concentrations and ionic strengths in the absence of polymeric crowding agent (*Figure 1—figure supplement 1*). From this, we focused our initial attention on a condition (25 µM ISB, 75 mM NaCl) that yields robust droplet formation (*Figure 1B*). Indeed, for subsequent HXMS experiments (described below), we sought to measure the behavior of an essentially homogenous droplet population since a highly heterogenous mixture of droplet and non-droplet ISB populations would likely yield convoluted mass spectra that would be challenging to properly assign to one of the states. In the condition we identified, 90%±5% of the ISB protein was found within the rapidly sedimenting droplet population (*Figure 1C*).

We designed an HXMS experiment to compare the polypeptide backbone dynamics of the ISB in the free and droplet states (*Figure 1D*). HXMS measures amide proton exchange, and for any generic protein, protection from HX is observed when secondary structures engage amide protons in hydrogen bonds (*Englander, 2006*). In our prior studies, we have utilized HXMS to readily identify contact points between domains of a multi-domain enzyme during its activation and inhibition (*Dawicki-McKenna et al., 2015*; *Zandarashvili et al., 2020*) as well as when components are added in a stepwise fashion during macromolecular complex assembly (*Falk et al., 2015*; *Guo et al., 2017*). We reasoned that ISB backbone dynamics would be restricted upon droplet formation, since the generally accepted broadscale basis of phase separation is through intermolecular interactions, albeit transient ones. In the case of ISB, we assumed that inter-heterotrimer interactions were the basis of its subsequent phase separation. HXMS is routinely performed over a time course, and we developed an approach to monitor droplet formation behavior of the samples alongside the HX reactions themselves (*Figure 1D*). Instead of letting the droplets settle on the slide, as in *Figure 1B*, we monitored them as they exist immediately upon preparing the slides for imaging to provide a rapid readout of droplet formation at each timepoint (*Figure 1E*). Robust droplet formation was observed in HX reaction conditions at all timepoints, including at the earliest one taken (10 s; *Figure 1E*). By the latest timepoint, 3000 s, there was some diminution in the number of droplets (*Figure 1E*), which may indicate the start of a transition of the droplets to a more solid state (i.e. gel-like). Thus, we concluded that timepoints longer than 3000 s would likely not be informative on how ISB backbone dynamics are impacted by initial droplet formation. This time course of HX proved to be sufficient to observe extensive exchange on all folded portions of the ISB, with the flexible regions lacking secondary structure exchanged much earlier (*Figure 2—figure supplements 1 and 2*). Slower HX was observed for all known and predicted secondary structural elements, except for the C-terminal helix of Borealin (*Figure 2—figure supplement 1*). Reciprocally, all predicted loop regions were very fast to exchange (i.e. essentially completely exchanged by 10 s), except for a small region around amino acids 140–150 of Borealin (*Figure 1F* and *Figure 2—figure supplement 3A–C*). This region was originally interpreted to be largely unstructured and contain high amounts of intrinsic disorder; however, our HXMS analysis suggests that some secondary structural elements exist in this region and are central to phase separation. Notably, this region overlaps with a deletion mutant (Borealin$^{\Delta 139-160}$) that causes a loss of phase separation (*Trivedi et al., 2019*).

To identify regions impacted by droplet formation, we first focused on an intermediate timepoint, 100 s, because visual inspection of HX patterns (*Figure 2—figure supplement 1*) indicated that there were clear changes at this point within the time course. At the 100 s timepoint, the most prominent differences between the soluble and droplet state were located within the three-helix bundle of the ISB, with long stretches in two subunits (INCENP and Borealin) and a small region at the N-terminal portion of the impacted α-helix in Survivin (*Figure 1F*). The only other region that corresponded to slower HX was within the aforementioned region of Borealin (amino acids 140–150), displaying rates consistent with the presence of secondary structure when the ISB is in its free state, which became further accentuated within the droplet state (*Figure 1F*). At the 300 s timepoint, a similar pattern is observed for the INCENP and Borealin proteins, with the notable addition of more extensive HX protection upon droplet formation within the three-helix bundle helix from Survivin and deprotection throughout its histone-recognizing BIR domain (*Figure 2—figure supplement 3A, B, D*). Taken together, the changes we observe in HX upon droplet formation indicates that discrete regions within structured portions of the ISB have slower backbone dynamics when in the droplet state.

## Two of three bundled ISB α-helices protected from HX in droplets

We focused on the three prominent regions of the interacting α-helices of INCENP and Borealin. HX protection within INCENP is strongest at the 100 s timepoint, especially within the C-terminal portion of the α-helix (*Figure 2A–C*). Examination of the entire time course shows that during intermediate levels of HX (i.e. between 100 and 1000 s), this region takes about three times as long to undergo the same amount of exchange when the ISB is in the droplet state relative to when it's in the free protein state (*Figure 2B and C* and *Figure 2—figure supplement 1*). Upon droplet formation, HX protection within Borealin is primarily located in the interacting α-helix and is less pronounced at any given peptide when compared to INCENP peptides (*Figure 2E*). Nonetheless, similar to INCENP peptides, it still takes about twice as long to achieve the same level of deuteration for this region of Borealin in the droplet state as compared to the free state (*Figure 2F and G* and *Figure 2—figure supplement 1*). In comparison, other regions exist within the ISB complex where multiple partially overlapping peptides show no measurable HX differences between droplet and free protein states (*Figure 2D*). This verifies that there are no properties of droplets, such as vastly different molar concentrations of $H_2O$ (or $D_2O$), that impact the general chemical exchange rate between all parts of the ISB. Rather, we conclude the changes observed within the ISB complex, such as those displayed in the two long interacting α-helices of INCENP and Borealin, are due to interactions formed between ISB complexes within the droplet state relative to those in the free state.

## Phase separation involves an acidic surface created by INCENP

We set to generate mutants to test the hypothesis that liquid-liquid demixing requires an interaction with the C-terminal portion of the long α-helix of INCENP within the three-helix bundle because this was the region with the greatest difference of HX protection between the droplet and free states (*Figure 2A*). We anticipated an electrostatic component to ISB phase separation since droplet formation is restricted at higher ionic strength (*Figure 1—figure supplement 1*). A conspicuous stretch of glutamic acid residues (E35, E36, E39, E40, and E42; *Figure 3A*) overlaps with a region of surface acidic charge in a single ISB heterotrimer (*Jeyaprakash et al., 2007*). We found that mutation of all five glutamic acid residues to alanine ($I_{Mut1}SB$; *Figure 3B*) to remove the acidic charge within the region caused a visible reduction in large droplets relative to $(ISB)_{WT}$ (*Figure 3C*), although it was difficult to measure a difference using a standard turbidity ($A_{330}$) measurement (*Figure 3D*). Mutation of the five glutamic acid residues to arginine ($I_{Mut2}SB$; *Figure 3B*) to reverse the charge yielded a predictably more pronounced effect, observed in both the microscope-based and turbidity assessments (*Figure 3C and D*). We conclude that some or all the five INCENP glutamic acid residues are involved in ISB phase separation.

## Crystal packing-guided mutagenesis to disrupt phase separation

Liquid-liquid phase separation has been long studied as involving on- or off-pathway nucleation of a crystal lattice, occurring in different parts of the same phase diagram (*Xu et al., 2021*). Therefore, we assessed the crystal packing of ISB heterotrimers, and found that INCENP and Borealin from separate ISB heterotrimers are in close contact (PDB#2QFA) (*Jeyaprakash et al., 2007*). Indeed, there were

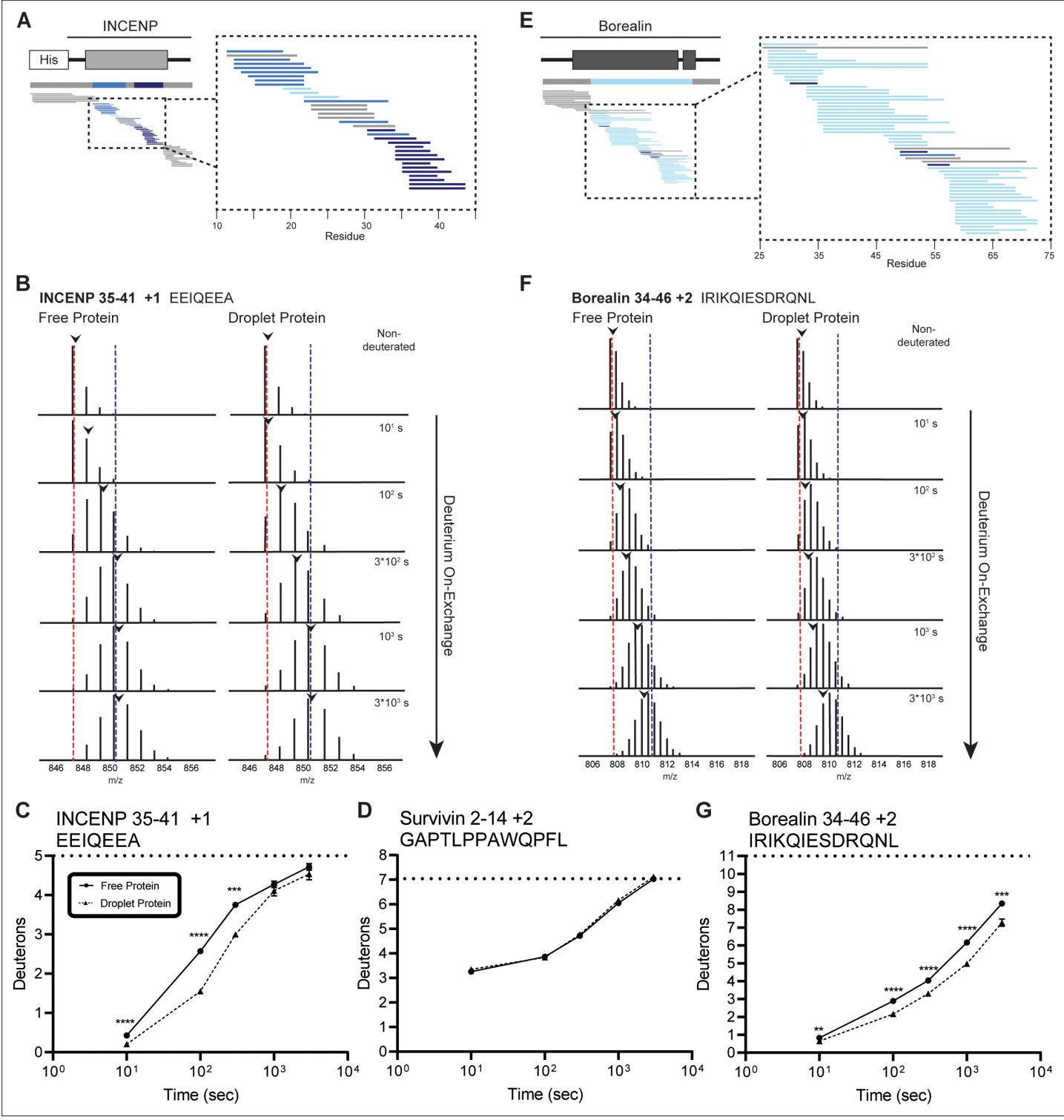

**Figure 2.** Regions of the three-helix bundle structure of INCENP and Borealin become protected from hydrogen/deuterium exchange (HX) upon phase separation. (**A**) Percent difference in HX upon phase separation at 100 s in the indicated region of INCENP. (**B**) Raw MS data of a representative peptide from indicated region of INCENP. Centroid values are indicated with an arrowhead. Red and blue dotted lines serve as guides for visualizing differences. The red line lies on mono-isotopic peak whereas the blue line lies on the centroid value for the largest timepoint (3000 s) within the free protein sample. (**C**) Hydrogen/deuterium exchange mass spectrometry (HXMS) of representative peptide from Panel B. The measured maximum number of exchangeable deuterons (maxD) when corrected with the average back exchange level (*Figure 2—figure supplement 2B*) is indicated. Data are represented as mean ± s.e.m.; note: the error is too small to visualize outside of readable data points except in one instance. Statistical analysis was

*Figure 2 continued on next page*

*Figure 2 continued*

performed using multiple unpaired t-tests. ****p<0.0001; ***0.0001<p<0.001; **0.001<p<0.01. (**D**) HXMS of a peptide from the indicated region within Survivin and displayed, as described in Panel C. This peptide shows the representative behavior of regions with the ISB that do not undergo changes in HX upon phase separation. Data are represented as mean ± s.e.m.; note: the error is too small to visualize outside of readable data points. (**E**) Percent difference in HX upon phase separation at 100 s in the indicated region of Borealin. (**F**) Raw MS data of a representative peptide from indicated region of Borealin. Centroid values are indicated with an arrowhead. Red and blue dotted lines serve as guides for visualizing differences, as explained in Panel B. (**G**) HXMS of representative peptide from Panel F and displayed as described in Panel C. Note: the error is too small to visualize outside of readable data points except in one instance.

The online version of this article includes the following source data, source code, and figure supplement(s) for figure 2:

**Source code 1.** R script to produce ribbon diagrams for each hydrogen/deuterium exchange mass spectrometry (HXMS) dataset.

**Source data 1.** Data used to generate *Figure 2*.

**Figure supplement 1.** Ribbon plots for free and droplet ISB protein.

**Figure supplement 2.** Extent of deuteration within fully deuterated (FD) hydrogen/deuterium exchange mass spectrometry (HXMS) control samples.

**Figure supplement 3.** Percent difference in hydrogen/deuterium exchange (HX) calculated for each peptide at 300 s.

**Figure supplement 3—source data 1.** Data used to generate *Figure 2—figure supplement 3*.

interactions between the HX protected regions of INCENP and Borealin (*Figure 2A and E*), including several side chains that we hypothesized to be involved in complementary electrostatic interactions (*Figure 4A*). Specifically, we noted three acidic INCENP residues mutated in $I_{Mut2}SB$ were within salt-bridge distance (i.e. ~2–4 Å) from a corresponding positive residue on Borealin (*Figure 3*). We used this as the basis for a second round of mutagenesis (*Figure 4B–D*). Mutation of two of the acidic INCENP

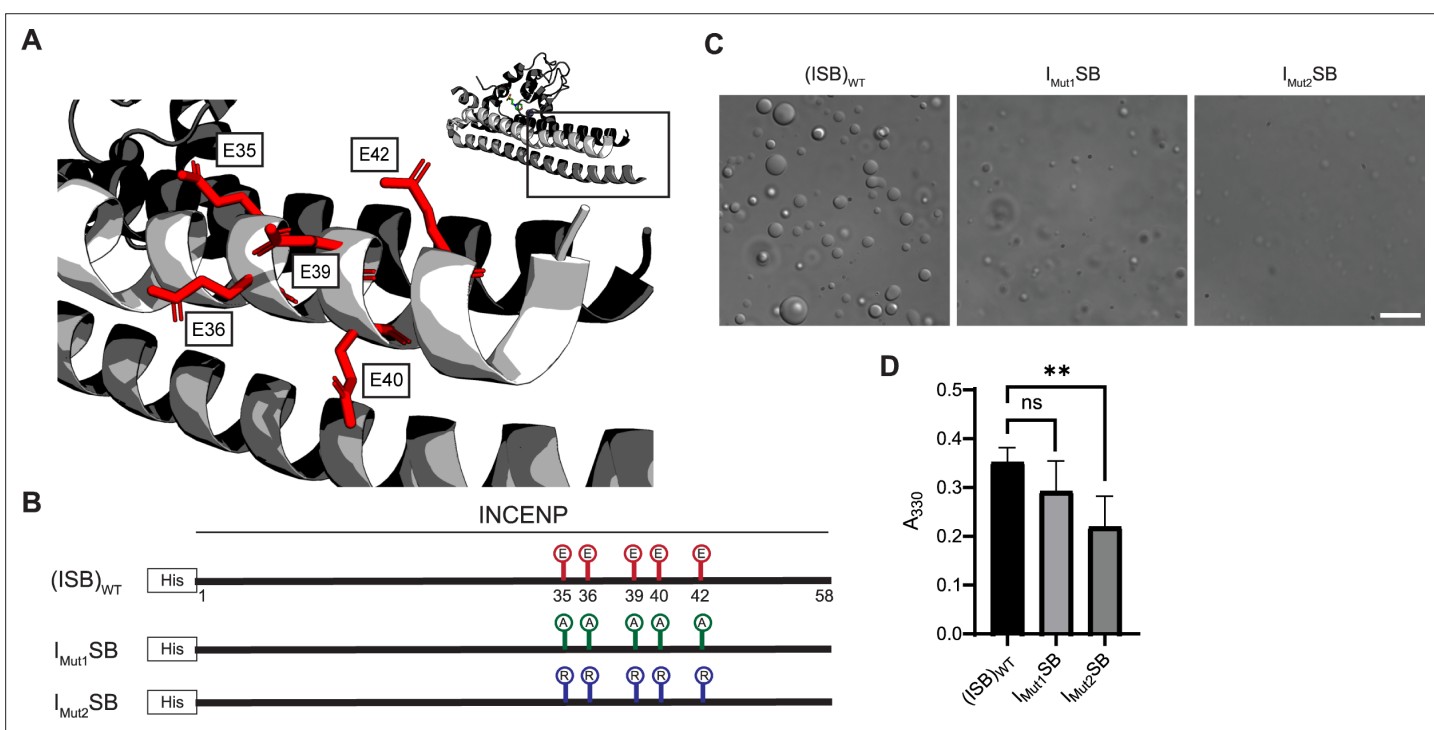

**Figure 3.** Acidic patch within INCENP coiled-coiled region contributes to electrostatic interaction within droplets. (**A**) Location of indicated acidic residues (E35/36/39/40/42) within INCENP at the surface of the coiled-coiled structure. Side chains are colored in red to indicate acidic charge. (**B**) Summary of a first round of mutations made to acidic residues within INCENP. Lolli-pop sticks represent each of the five residues in question. For $(ISB)_{WT}$, red color indicates acidic charge. For $I_{Mut1}SB$, green color indicates neutral charge. For $I_{Mut2}SB$, blue color represents basic charge. (**C**) DIC micrographs of the ISB droplets for $I_{Mut1}SB$ and $I_{Mut2}SB$. The micrograph for $(ISB)_{WT}$ is from the same sample used in *Figure 1B*. Scale bar = 10 μm. (**D**) Turbidity calculations of $I_{Mut1}SB$ and $I_{Mut2}SB$ measured as absorbance at 330 nm; n=6 for $(ISB)_{WT}$, $I_{Mut1}SB$, and $I_{Mut2}SB$. Statistical analysis was performed using a Brown-Forsythe and Welch ANOVA test. **0.001<p<0.01.

The online version of this article includes the following source data for figure 3:

**Source data 1.** Data used to generate *Figure 3*.

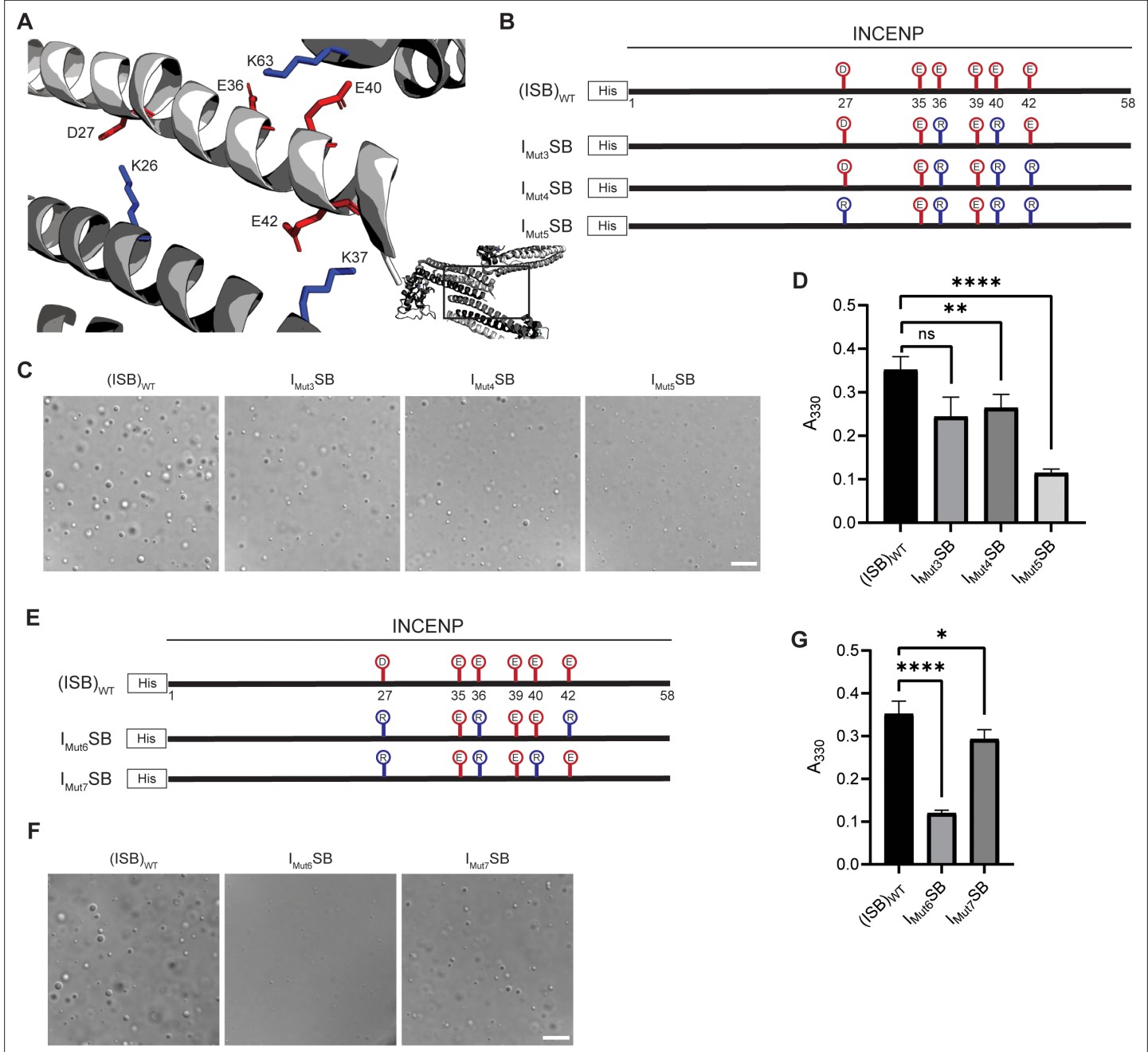

**Figure 4.** Crystal packing of ISB three-helix bundle structure highlights possible salt-bridges between multiple complexes. (**A**) Location of acidic and basic residues within crystal packing of ISB between INCENP$_1$ and Borealin$_2$/Borealin$_3$. Side chains are colored in red to indicate acidic charge and blue to indicate basic charge. (**B**) Summary of a second round of mutations made to acidic residues within INCENP. Lolli-pop sticks represent each of the indicated residues in question. (**C**) DIC micrographs of the ISB droplets for I$_{Mut3}$SB, I$_{Mut4}$SB, and I$_{Mut5}$SB. Scale bar = 10 μm. (**D**) Turbidity calculations of I$_{Mut3}$SB, I$_{Mut4}$SB, and I$_{Mut5}$SB measured as absorbance at 330 nm; n=6 for (ISB)$_{WT}$, I$_{Mut4}$SB, and I$_{Mut5}$SB. n=3 for I$_{Mut3}$SB. Statistical analysis was performed using a Brown-Forsythe and Welch ANOVA test. ****p<0.0001; **0.001<p<0.01. (**E**) Summary of a third round of mutations made to acidic residues within INCENP. Lolli-pop sticks represent each of the indicated residues in question. (**F**) DIC micrographs of the ISB droplets for I$_{Mut6}$SB and I$_{Mut7}$SB. Scale bar = 10 μm. (**G**) Turbidity calculations of I$_{Mut6}$SB and I$_{Mut7}$SB measured as absorbance at 330 nm; n=6 for (ISB)$_{WT}$. n=3 for I$_{Mut6}$SB and I$_{Mut7}$SB. Statistical analysis was performed using a Brown-Forsythe and Welch ANOVA test. ****p<0.0001; *0.01<p<0.05.

The online version of this article includes the following source data and figure supplement(s) for figure 4:

**Source data 1.** Data used to generate *Figure 4*.

**Figure supplement 1.** Highlighting structure of conflicting salt-bridge between INCENP and Borealin.

residues (E36 and E40; $I_{Mut3}SB$) led to a similar level of reduction in phase separation (*Figure 4C and D*) as when all five of the original glutamic acidic residues were mutated (*Figure 3*). The addition of either one (E36, E40, and E42; $I_{Mut4}SB$) or two (D27, E36, E40, and E42; $I_{Mut5}SB$) mutated acidic INCENP residues displayed either similar or increased levels of reduction, respectively (*Figure 4C and D*). These findings, along with our HX measurements (*Figure 1*), raised the possibility of shared interaction sites between liquid-liquid demixed and crystal forms of the ISB.

As a first test of this notion, we designed a third round of mutagenesis to probe any potential interactions of the acidic INCENP residues facing two different ISB trimers in the crystal lattice: INCENP$^{D27,E42}$ contact the Borealin subunit of one heterotrimer, while INCENP$^{E36,E40}$ contact another (*Figure 4A*). We predicted that the contacts responsible for phase separation closely correspond to the highlighted crystal contacts. If so, then those within typical bond distances (i.e. ~3–4 Å) would have a larger impact than those with distances in the crystal lattice that are too large to generate a salt-bridge. Along with mutation of INCENP$^{D27,E36}$, $I_{Mut6}SB$ and $I_{Mut7}SB$ vary by either including a mutation of INCENP$^{E42}$ (within salt-bridge distance [2.9 Å] of Borealin$^{K37}$; $I_{Mut6}SB$; *Figure 4—figure supplement 1*) or a mutation of INCENP$^{E40}$ (at too large a distance to bond with Borealin$^{K63}$; $I_{Mut7}SB$; *Figure 4—figure supplement 1*) (*Figure 4E*). We found that $I_{Mut6}SB$, wherein all three mutations impact salt-bridges, profoundly reduces phase separation (*Figure 4F and G*). On the other hand, $I_{Mut7}SB$ only has a minor effect on phase separation (*Figure 4F and G*). With the prior finding with $I_{Mut3}SB$, we deduce that INCENP$^{E36}$, but not INCENP$^{E40}$, contributes to phase separation. Combining the information from the three rounds of mutagenesis, we conclude that residues D27, E36, and E42 of INCENP contribute additively to the disruption of phase separation we observe within $I_{Mut6}SB$. Together, these findings provided an early indication that precise salt-bridges between ISB heterotrimers are key to its phase separation.

## Breaking and reforming salt-bridges to modulate ISB phase separation

We set to test our prediction that multiple salt-bridges between ISB heterotrimers drive its phase separation by breaking the salt-bridges from the opposite subunit (Borealin) and by reconstituting the salt-bridges through pairing each charge switch mutation with each other. A mutation to the Borealin subunit that contains the three relevant lysine to glutamic acid substitutions was designed based on the structural model and combined with either wild-type INCENP ($ISB_{Mut}$) or the Mut$_6$ version of INCENP ($I_{Mut6}SB_{Mut}$) (*Figure 5A and B*). Consistent with our prediction, $ISB_{Mut}$ was severely crippled in its ability to undergo phase separation (*Figure 5C–E*). Strikingly, the compensatory mutations in INCENP that reconstitute the salt-bridges between ISB hetrotetramers completely restore droplet formation in $I_{Mut6}SB_{Mut}$, detectable by microscopy and spectroscopy (*Figure 5C and D*), and almost entirely restores wild-type behavior, as measured by the ISB concentration required to saturate droplet formation (*Figure 5E* and *Figure 5—figure supplements 1 and 2*). We emphasize that the key mutants that break (including $I_{mut6}SB$ and $ISB_{mut}$) and re-form ($I_{mut6}SB_{mut}$) the salt-bridge-mediated droplet formation underwent three different measurements (turbidity [*Figures 4 and 5*], microscope-based detection [*Figures 4 and 5*], and sedimentation-based determination of saturation concentration [*Figure 5* and *Figure 5—figure supplement 1*]). These different types of measurements vary by several minutes due to practical considerations relative to the time from the initial reaction assembly. Thus, our conclusions about the phase separation of mutant versions of ISB heterotrimers are based on independent experiments that span the time window covered in our initial HXMS analysis of $ISB_{WT}$ (*Figures 1 and 2*). The potent rescue when the two surfaces are simultaneously mutated provides clear support for the conclusion that salt-bridging between these parts of the structured portions of ISB drive phase separation.

## Mutation of Borealin to disrupt salt-bridges reduces phase separation in cells

To test whether or not the inter-CPC salt-bridges we identified can impact phase separation in cells, we employed the Cry2 optoDroplet system (*Shin et al., 2017*), comparing Borealin$_{WT}$ to Borealin$_{Mut}$ (i.e. the same mutations present in $ISB_{Mut}$ in *Figure 5*). In this system, Borealin fused to Cry2, a light-inducible dimerizing protein, and mCherry (for fluorescent detection) readily forms droplets after exposing cells to blue light (*Trivedi et al., 2019*). Importantly, these droplets are light dependent, form in the nucleus, and recruit endogenous Aurora B (*Figure 6A* and *Figure 6—figure supplements 1 and 2*).

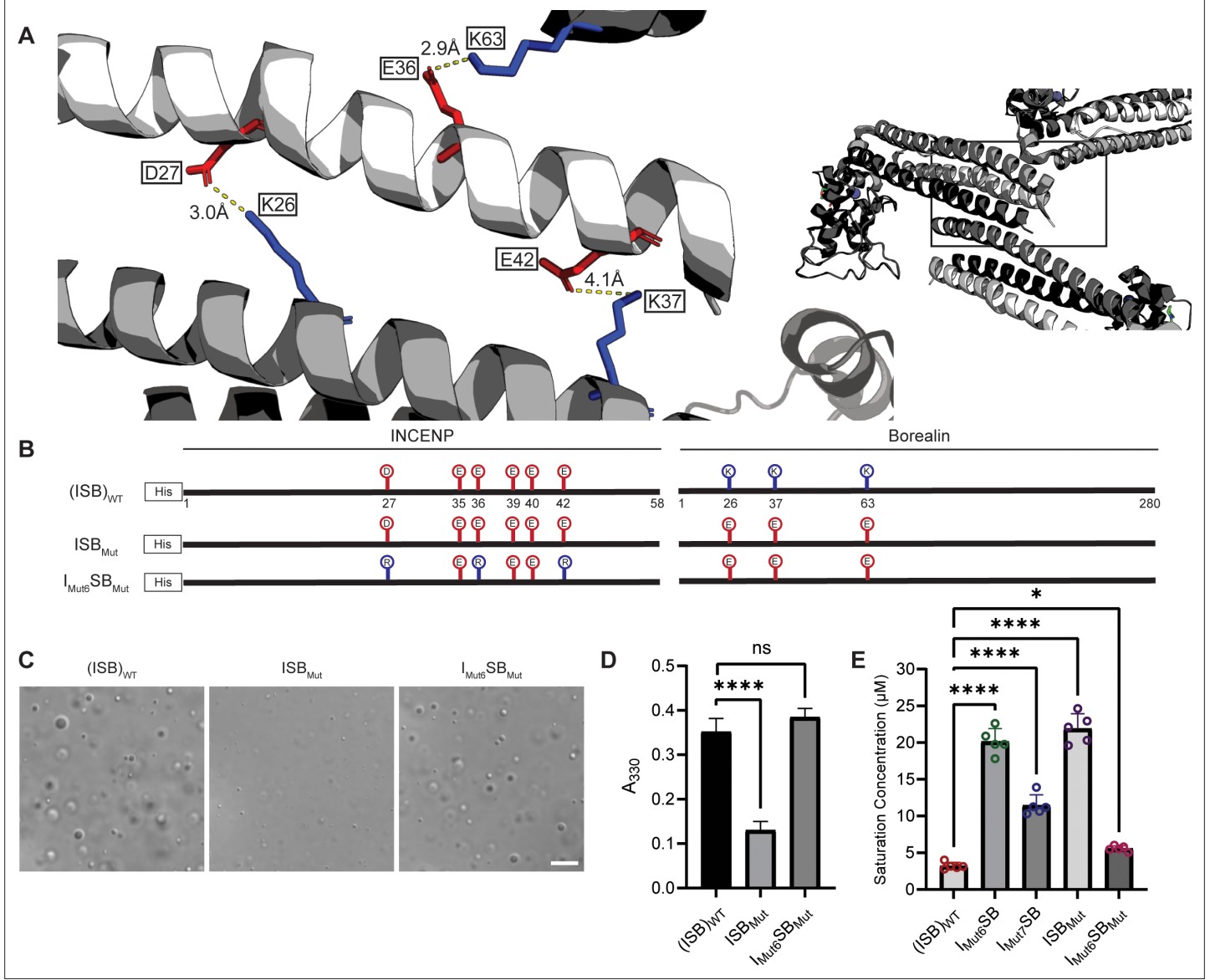

**Figure 5.** Salt-bridges between multiple ISB complexes provide multivalency required for phase separation. (**A**) Location of key salt-bridges within crystal packing of ISB between INCENP$_1$ and Borealin$_2$/Borealin$_3$. Side chains are colored in red to indicate acidic charge and blue to indicate basic charge. Distances between side chains are indicated. (**B**) Summary of a fourth round of mutations made to acidic residues within INCENP and basic residues within Borealin. Lolli-pop sticks represent each of the indicated residues in question. (**C**) DIC micrographs of the ISB droplets for ISB$_{Mut}$ and I$_{Mut6}$SB$_{Mut}$. Scale bar = 10 µm. (**D**) Turbidity calculations of ISB$_{Mut}$ and I$_{Mut6}$SB$_{Mut}$ measured as absorbance at 330 nm; n=6 for (ISB)$_{WT}$. n=3 for ISB$_{Mut}$ and I$_{Mut6}$SB$_{Mut}$. Statistical analysis was performed using a Brown-Forsythe and Welch ANOVA test. ****p<0.0001. (**E**) Saturation concentration of (ISB)$_{WT}$, I$_{Mut6}$SB, I$_{Mut7}$SB, ISB$_{Mut}$, and I$_{Mut6}$SB$_{Mut}$ in buffer containing 75 mM NaCl measured using sedimentation. n=5 for (ISB)$_{WT}$, I$_{Mut6}$SB, I$_{Mut7}$SB, ISB$_{Mut}$, and I$_{Mut6}$SB$_{Mut}$. Statistical analysis was performed using a one-way ANOVA test with Dunnett's multiple comparisons test. ****p<0.0001; *0.01<p<0.05.

The online version of this article includes the following source data and figure supplement(s) for figure 5:

**Source data 1.** Data used to generate *Figure 5*.

**Figure supplement 1.** SDS-PAGE gels from saturation concentration experiment.

**Figure supplement 1—source data 1.** Data used to generate *Figure 5—figure supplement 1*.

**Figure supplement 2.** ISB-WT and mutant protein complexes (SDS-PAGE).

**Figure supplement 2—source data 1.** Data used to generate *Figure 5—figure supplement 2*.

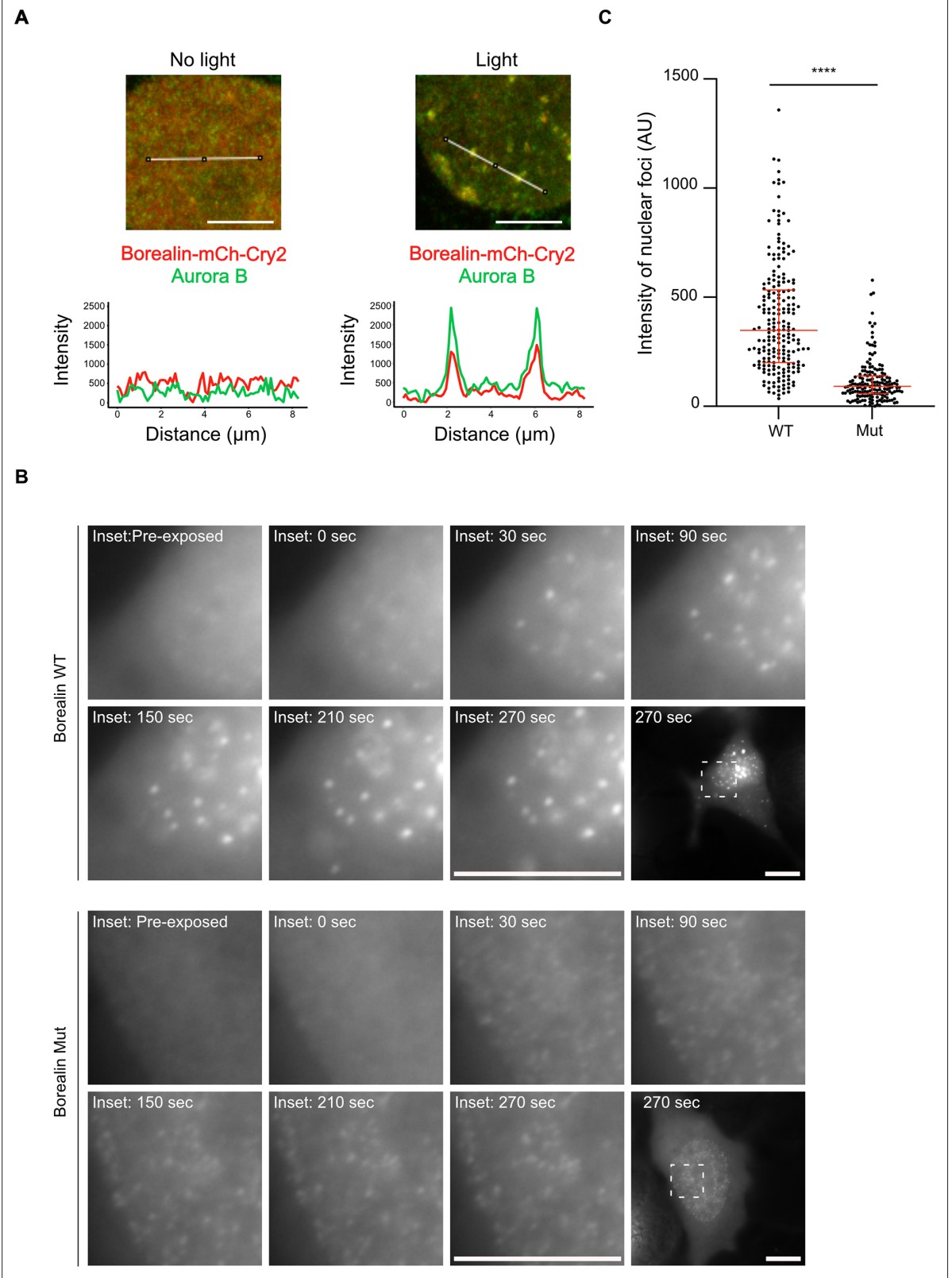

**Figure 6.** Disrupting salt-bridge residues in Borealin diminishes phase separation in cells. (**A**) Endogenous Aurora B is recruited to nuclear Borealin foci upon exposure to 488 nm light. Images within the nucleus are shown (see *Figure 6—figure supplement 1* for images of the entire cell for each of these enlarged views). The positions of the line scans (below) are indicated in the images by a white line. Scale bar = 5 μm. (**B**) Fluorescent detection of Borealin$_{WT}$ or Borealin$_{Mut}$, each fused to mCherry-Cry2 in an optoDroplet assay. Images were collected before and after (at the indicated timepoints)

*Figure 6 continued on next page*

*Figure 6 continued*

exposure to 488 nm light to induce Cry2 dimerization (note that the images were acquired with the same imaging conditions and scaled in the same manner for display). Scale bar = 10 μm. (**C**) Quantification of the intensity of foci. n=2 experiments, and 18 (WT) and 16 (Mut) cells. The results of an unpaired, non-parametric t-test, Mann-Whitney test is shown, wherein **** equates to a p-value <0.0001. The lines represent the median and the interquartile range.

The online version of this article includes the following figure supplement(s) for figure 6:

**Figure supplement 1.** Endogenous Aurora B is recruited to Borealin-mCherry-Cry2 droplets in the nucleus upon exposure to white light.

**Figure supplement 2.** Evidence supporting the engagement of the Borealin-mCherry-Cry2 with the endogenous chromosome passenger complex (CPC).

Since Aurora B and Borealin are indirectly linked through the INCENP subunit, these findings suggest that the entire CPC is engaged in the nuclear droplets. Furthermore, the Borealin-mCherry-Cry2 fusion protein is recruited to inner centromeres in mitosis (*Figure 6—figure supplement 2B*). Using this system, we find that while Borealin$_{Mut}$ can form droplets, the intensity of the mCherry signal in the droplets formed in the nucleus by Borealin$_{Mut}$ is less than that observed with Borealin$_{WT}$ (*Figure 6B and C*). These measurements indicate that mutating the salt-bridging residues we identified in Borealin by HXMS complimentarily reduces phase separation in the cellular environment.

## Discussion

We demonstrate the feasibility of using HXMS as an unbiased approach to map the protein-protein interactions that drive liquid-liquid demixing. In the absence of structural information or useful structural models (i.e. of multimeric complexes where current structural predictions fall short), HXMS provides localization information at moderately high resolution (i.e. within small numbers of amino acid residues given the coverage we achieved with the ISB, for instance). We envision that HXMS will be broadly useful to advance our physical understanding of the protein/protein and protein/nucleic acid interactions that drive phase separation and the formation of membraneless compartments within the cell.

It can be extremely difficult to generate structural information about the weak interactions that drive liquid-liquid demixing, but by combining HXMS with the long-studied relationship between liquid-liquid phase separation and crystal formation, our study shows that this can be done. Our approach might be generalizable for many proteins that have phase separation activities. Of course, packing information is largely thrown out during the presentation in most macromolecular structure-focused studies, but the data are readily available (i.e. archived in the PDB). We anticipate that clues from crystal contacts will help uncover potential sites that participate in other phase separating protein complexes, but they are unlikely to be sufficient. Rather, we envision HXMS as an essential step to localize the key contacts in the liquid-liquid demixed state, then most powerfully combined with structural information, including crystal contacts, when they are available.

Multivalency has emerged as the key driver of the liquid-liquid demixing of proteins and the interactions between monomeric units, and often involve weak interactions that are separated by intrinsically disordered domains. To date, detailed insight into the specific self-self interactions that are proposed to drive monomeric proteins and protein complexes into phase-separated compartments has been rare, despite an enormous and growing list of candidate compartments at diverse locations in the cell (*Banani et al., 2017*). Undoubtedly, all sorts of macromolecular interaction types will be utilized within this diverse collection of compartments. Our findings support the notion that structured domains can play major roles as drivers of liquid-liquid demixing. Specifically, we map two structured regions that are separated by an intrinsically disordered region as underlying the liquid-liquid demixing of a subcomplex of the CPC. Our data suggest that salt-bridges between highly ordered regions of one subunit form with an adjacent complex in a liquid-liquid demixed droplet. These salt-bridges can explain the low affinity, salt sensitivity, and transient nature of the liquid demixed state. Our data also provide an important clue about the previously identified region on Borealin that is required for liquid demixing in vitro and proper CPC assembly in cells (*Trivedi et al., 2019*). Specifically, our data (*Figure 1F* and *Figure 2—figure supplements 1 and 3A*) suggest this region of Borealin adopts secondary structure that undergoes additional HX protection in the liquid-liquid demixed state. We presume this region, enriched in basic residues, interacts with a negatively charged region

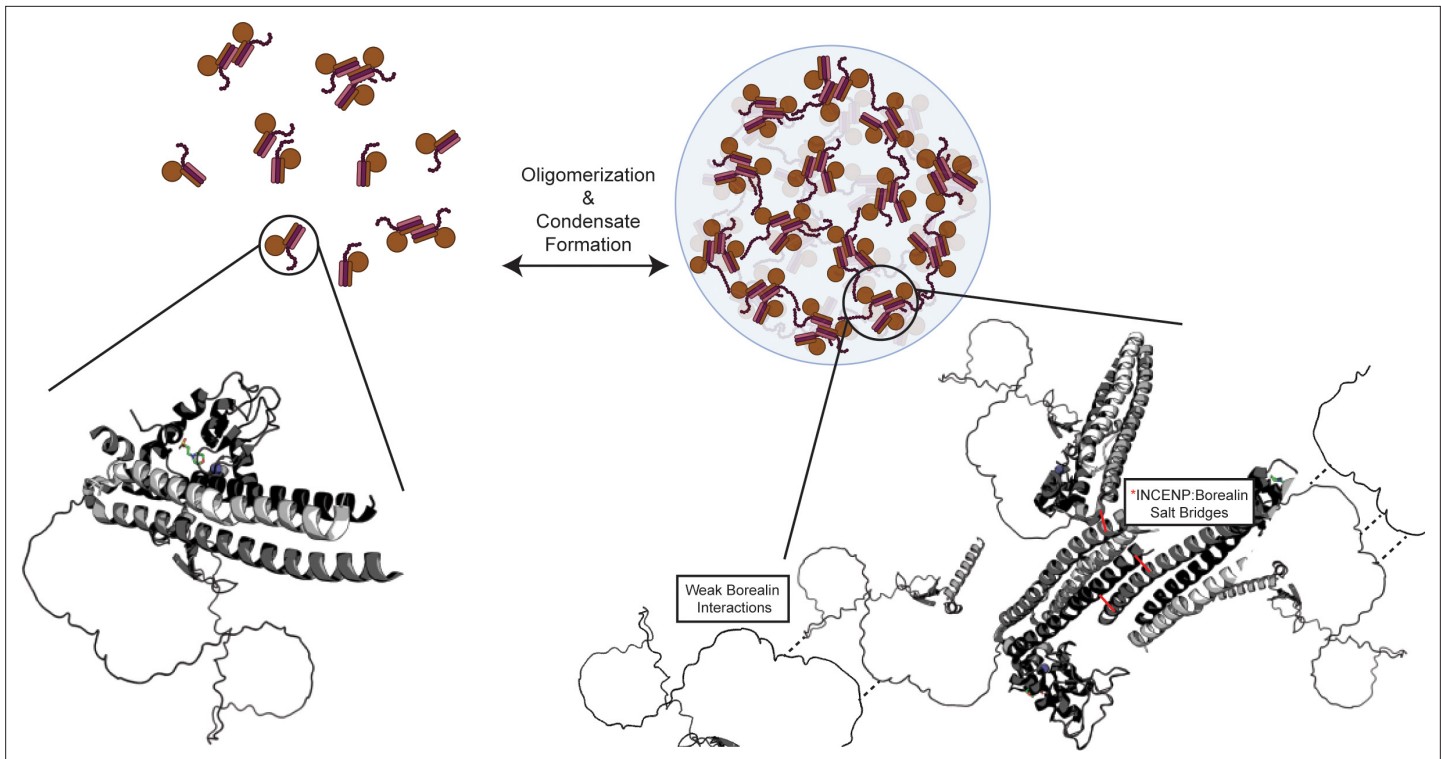

**Figure 7.** Summary model highlighting functionality of chromosome passenger complex (CPC) phase separation in cells. Stabilized interactions defined by hydrogen/deuterium exchange mass spectrometry (HXMS) findings are indicated in solid black lines, while proposed weak interactions via Borealin loop are defined by dashed line. This model utilizes an AlphaFold prediction for the unstructured region of Borealin (*Jumper et al., 2021*; *Varadi et al., 2022*).

that remains to be identified in an adjacent heterotrimer. We note that there is a region of strong negative charge within the intrinsically disordered region between the two regions of HX protection on the Borealin subunit. Our findings integrate into the emerging concept of 'stickers and spacers' (*Mittag and Pappu, 2022*), where the driving stickers for the CPC are indeed the specific salt-bridges that we identified through a combination of HXMS, analysis of ISB crystal packing, and mutagenesis. The spacers include the regions that lack stable secondary structure (i.e. the substantial portions where HX is essentially complete by the 10 s timepoint; *Figure 2—figure supplement 1*). The stickers are sufficient for higher-order assembly, and the spacers permit particular solvation properties that contribute to the nature of the phase separation properties (*Aguzzi and Altmeyer, 2016*; *Choi et al., 2020*; *Gomes and Shorter, 2019*; *Mittag and Pappu, 2022*) and downstream consequences to the viscoelastic properties of the inner centromere compartment (*Figure 7*).

Inner centromere formation is highly regulated so that the non-membranous compartment only forms during prophase and can be quickly disassembled at the beginning of anaphase (*Trivedi and Stukenberg, 2016*). The reaction requires a combination of specific recruitment of the CPC to histone posttranslational modifications (*Kelly et al., 2010*; *Wang et al., 2010*; *Yamagishi et al., 2010*), interactions of the Borealin subunit with histones, and interactions between Borealin and the Sgo1 protein, which in turn is recruited by histone phosphorylation in addition to the multivalent interactions between the INCENP, Survivin, and Borealin subunits that are the subject of this work. Although we initially suggested that recruitment relies upon initial recognition of H3[T3phos] directly by Survivin (*Wang et al., 2010*) – and indirectly by interaction with the Sgo2 adaptor protein that recognizes H2A[T120phos] (*Yamagishi et al., 2010*) – which then drives liquid demixing, it is also possible that the system is so finely tuned that the CPC can both interact with phosphohistones and also interact with other CPCs by multivalency, generating a chromosome-localized compartment capable of efficiently mediating

robust mitotic error correction. Furthermore, a very recent report indicates that CPC phase separation can be modulated by MLL1-mediated methylation of the Borealin subunit (*Sha et al., 2023*).

It has long been appreciated that Aurora B, the catalytic subunit of the CPC, is a prime example of a kinase whose activity is dictated by localization (*Carmena et al., 2012*). We focus on the inner centromere, where Aurora B monitors connections of chromosomes to the mitotic spindle, and it is also vital later in mitosis for the abscission checkpoint and the regulation of cytokinesis (*Barr and Gruneberg, 2007*). Since perturbing the ability of the CPC to phase-separate at the inner centromere leads to defects in mitosis (*Trivedi et al., 2019*; *Trivedi and Stukenberg, 2020*), the role of phase separation appears to be tuning levels at the centromere, producing a functional inner centromere compartment, and maintaining a pool of the CPC after chromosome alignment destined for down-stream steps in cytokinesis. In the present study, we have defined the impact of phase separation on the structural dynamics of the responsible regions for separation of the CPC, and, further, identified the key structural determinants that drive the inter-heterotrimer interactions. Recapitulating the environment of the inner centromere with purified components will constitute a substantial future challenge for the field, and our work provides a framework with which to understand this critical region of mitotic chromatin and a powerful experimental approach, HXMS, with which to probe the protein components of the inner centromere.

## Materials and methods

### Key resources table

| Reagent type (species) or resource | Designation | Source or reference | Identifiers | Additional information |
|---|---|---|---|---|
| Sequence-based reagent | WTISB_F | This Paper | PCR primer | 3' - TGAGATCCGAATTCGAGCTCTAATTTTG - 5' |
| Sequence-based reagent | WTISB_R | This paper | PCR primer | 3' - GCTGTGATGATGATGATGATGGCTGCTG - 5' |
| Sequence-based reagent | ISBMut6_F | This paper | PCR primer | 3' - CTTGAGCGTATCCAAGAGGAGGCCCG ACGCATGTTCACC - 5' |
| Sequence-based reagent | ISBMut6_R | This paper | PCR primer | 3' - GGTGAACATGCGTCGGGCCTCCTCT TGGATACGCTCAAG - 5' |
| Sequence-based reagent | ISBMut7_F | This paper | PCR primer | 3' - CGTATCCAAGAGCGAGCCGAGCGCA TGTTCACCAGAGAA - 5' |
| Sequence-based reagent | ISBMut7_R | This paper | PCR primer | 3' - TTCTCTGGTGAACATGCGCTCGGC TCGCTCTTGGATACG - 5' |
| Sequence-based reagent | WTISB_F_2 | This paper | PCR primer | 3' - CCGTCTCGCCCAAATCTGCA - 5' |
| Sequence-based reagent | WTISB_R_2 | This paper | PCR primer | 3' - GCTGTGATGATGATGATGAT GGCTGCTG - 5' |
| Sequence-based reagent | $I_{Mut1}SB\_G\_Block$ | This paper | Oligonucleotide | See *Supplementary file 2* |
| Sequence-based reagent | $I_{Mut2}SB\_G\_Block$ | This paper | Oligonucleotide | See *Supplementary file 2* |
| Sequence-based reagent | $I_{Mut3}SB\_G\_Block$ | This paper | Oligonucleotide | See *Supplementary file 2* |
| Sequence-based reagent | $I_{Mut4}SB\_G\_Block$ | This paper | Oligonucleotide | See *Supplementary file 2* |
| Sequence-based reagent | $I_{Mut5}SB\_G\_Block$ | This paper | Oligonucleotide | See *Supplementary file 2* |
| Sequence-based reagent | $ISB_{Mut\_}G\_Block$ | This paper | Oligonucleotide | See *Supplementary file 2* |
| Sequence-based reagent | $I_{Mut6}SB_{Mut\_}G\_Block$ | This paper | Oligonucleotide | See *Supplementary file 2* |
| Strain, strain background (*Escherichia coli*) | Rosetta 2 (DE3) plysS | Novagen | 71403 | Electrocompetent cells |
| Cell line (*Homo sapiens*) | T-Rex HeLa Cell Line | Thermo Fisher Scientific | R71407 | |
| Recombinant DNA reagent | pET28a_ISB | *Trivedi et al., 2019* | | 6xHis-INCENP[1-58], FL Survivin and FL Borealin |
| Commercial assay, kit | NEB Hifi DNA Assembly Kit | New England Biolabs | E5520S | For molecular cloning |
| Other | HisTrap HP Column | Cytiva/GE Life Sciences | 17524801 | For protein purification |
| Other | Hi-Load 16/60 Superdex-200 pg | Cytiva/GE Life Sciences | 28989335 | For protein purification |

*Continued on next page*

*Continued*

| Reagent type (species) or resource | Designation | Source or reference | Identifiers | Additional information |
|---|---|---|---|---|
| Other | C18 HPLC Column, 0.3×75 mm$^2$ | Agilent | | For HXMS experimentation |
| Other | TARGA C8 5 µM Piccolo HPLC column | Higgins Analytical | | For HXMS experimentation |
| Other | Leica DMI6000 B | Leica Microsystems | | For differential interference contrast microscopy |
| Other | Discovery M120SE Sorvall Ultracentrifuge | New Life Scientific | | For sedimentation and saturation concentration assays |
| Other | LTQ Orbitrap XL Mass Spectrometer | Thermo Fisher Scientific | | For HXMS data acquisition |
| Other | Exactive Plus EMR Orbitrap Mass Spectrometer | Thermo Fisher Scientific | | For HXMS data acquisition |
| Other | NanoDrop 2000 UV-Vis Spectrophotometer | Thermo Fisher Scientific | ND2000CLAPTOP | For turbidity measurements |
| Other | Zeiss Observer-Z1 Microscope | Zeiss | | For optoDroplet assay |
| Software | XCalibur | Thermo Fisher Scientific | OPTON-30965 | For HXMS data acquisition |
| Software | ExMS2 | *Kan et al., 2019* | | For HXMS data processing |
| Software | MATLAB | Mathworks | | For HXMS data processing |
| Software | RStudio | Posit | | For HXMS data processing |
| Software | Bioworks 3.3 | Thermo Fisher Scientific | | For HXMS data processing |
| Software | HDExaminer | Sierra Analytics | | For HXMS data processing |
| Software | GelQuant Express Analysis Software | Fisher Scientific | | For densitometry measurements |
| Software | Fiji (ImageJ) | National Institutes of Health (NIH) | | To analyze images |
| Software | Prism | GraphPad | | For data processing |

## Protein purification

Rosetta 2 (DE3) pLysS cells were transformed with a triscistronic pET28a vector containing sequences for 6xHis-INCENP[1-58], full-length survivin, and full-length Borealin. Cells were then grown in the presence of 30 µg/ml kanamycin to an optical density between 0.6 and 0.8 and protein expression was induced with 1 mM isopropylthiogalactoside for 16–18 hr at 18°C. The medium was also supplemented with 60 mg/l ZnCl$_2$ and 0.2% glucose. Cells were then pelleted and lysed in buffer containing 50 mM Tris pH 7.5, 500 mM NaCl, 5% glycerol, 5 mM imidazole, 5 mM 2-mercaptoethanol (BME) and protease inhibitor cocktail (Roche) using a combination of Dounce homogenization and sonication. The lysate was then cleared by centrifugation and purified over HisTrap HP column (Cytiva) and eluted using 50 mM Tris pH 7.5, 500 mM NaCl, 5% glycerol, 500 mM imidazole, 5 mM BME at 4°C. The elutate was further gel-filtered over a Hi-Load 16/60 Superdex-200 pg column (GE Life Sciences, Cytiva) in buffer containing 50 mM Tris pH 7.5, 500 mM NaCl, 5% glycerol, and 5 mM BME. The desired fractions were collected and concentrated using Amicon Ultra-4 Centrifugal Filter Units with 3 kDa cut-off. All mutants within this study are purified similarly (*Figure 5—figure supplement 2*).

## Plasmid construction and mutagenesis

The I$_{Mut1}$SB, I$_{Mut2}$SB, I$_{Mut3}$SB, I$_{Mut4}$SB, and I$_{Mut5}$SB constructs were created by a two-fragment assembly system (NEB), replacing WT INCENP[1-58] sequence with the corresponding gBlock Gene Fragments (IDT). WT template DNA was amplified via PCR (Forward Primer: WTISB_F, Reverse Primer: WTISB_R) before assembly. The I$_{Mut6}$SB construct was created by the Quikchange protocol (Stratagene) (Forward Primer: ISBMut6_F, Reverse Primer: ISBMut6_R), using the I$_{Mut5}$SB construct as the template DNA. The I$_{Mut7}$SB construct was created by Quikchange protocol (Stratagene) (Forward Primer: ISBMut7_F, Reverse Primer: ISBMut7_R), using the I$_{Mut5}$SB construct as the template DNA. The ISB$_{Mut}$ and I$_{Mut6}$SB$_{Mut}$ constructs were created by a two-fragment assembly system (NEB), replacing WT INCENP[1-58] and WT

Borealin sequence with the corresponding gBlock Gene Fragment (IDT). WT template DNA was amplified via PCR (Forward Primer: WTISB_F_2, Reverse Primer: WTISB_R_2) before assembly. Sequences were verified by automated cycle sequencing (University of Pennsylvania Genomics Analysis Core).

## Phase separation assay

Phase separation was induced by diluting the indicated amount of ISB in a low salt buffer (50 mM Tris pH 7.5 and 5 mM BME) to achieve the indicated final concentration of protein and NaCl (25 µM ISB, 75 mM NaCl). Protein was always added last to each reaction. Phase separation was observed by adding a small volume of the reaction onto a coverslip and then imaging the ISB droplet by differential interference contrast (DIC) microscopy. All movies and images were captured within 5 min of the reaction setup. For time-lapse imaging of ISB droplet fusion, ISB droplets were formed in the indicated conditions and immediately imaged via DIC every second. Imaging of the ISB droplet during HXMS experimentation was captured as close to the indicated timepoint as possible.

## Sedimentation assay

Following liquid-liquid phase separation, each reaction was allowed to stand for 100 s and then centrifuged at 16,100 × $g$ for 10 min to separate the soluble phase from the droplet phase. All of the top phase was removed and placed in a separate tube. The dense phase was resuspended in an equivalent volume of purification buffer (50 mM Tris pH 7.5, 500 mM NaCl, 5% glycerol, 5 mM BME). Then, 10 µL from each top phase and dense phase was removed and analyzed using SDS-PAGE.

## HXMS measurement and analysis

Deuterium on-exchange of soluble ISB protein was performed at room temperature (25°C) by diluting purified ISB with deuterium on-exchange buffer (50 mM Tris pD 7.5, 500 mM NaCl) to a final protein concentration of 25 µM ISB, 500 mM NaCl, and a final $D_2O$ content of 75%. A 20 µL aliquot was removed at each timepoint (10, 100, 300, 1000, 3000 s) and the reaction was quenched with 30 µL ice-cold quench buffer (1.67 M guanidine hydrochloride, 8% glycerol, and 0.8% formic acid, for a final pH of 2.4–2.6) and rapidly frozen in liquid nitrogen. The samples were stored at –80°C until analysis by MS. Deuterium on-exchange of phase-separated ISB protein was performed at a similar temperature by diluting purified ISB with a mixture of two on-exchange buffers (Buffer 1: 50 mM Tris pD 7.5, 0 mM NaCl; Buffer 2: 50 mM Tris pD 7.5, 500 mM NaCl) to a final protein concentration of 25 µM ISB, 75 mM NaCl, and a final $D_2O$ content of 75%. pD values are direct pH meter readings. Samples were prepared and frozen in a similar manner to soluble ISB protein. All samples were produced in quadruplicate so that there would be a spare in addition to a triplicate set to measure, in case of a technical issue in downstream steps. The supplementary table (see *Supplementary file 1*) summarizes the HXMS experiments.

HX samples were individually thawed at 0°C for 2.5 min, then injected (50 µL) and pumped through an immobilized pepsin (Sigma) column at an initial flow rate of 50 µL/min for 2 min followed by 150 µL/min for 2 min. Pepsin was immobilized by coupling to POROS 20 AL support (Applied Biosystems) and packed into column housings of 2 mm × 2 cm (64 µL) (Upchurch). Protease-generated fragments were collected onto a TARGA C8 5 µM Piccolo HPLC column (1.0×5.0 mm$^2$, Higgins Analytical) and eluted through an analytical C18 HPLC column (0.3×75 mm$^2$, Agilent) by a shaped 12–100% Buffer B gradient over 25 min at 6 µL/min (Buffer A: 0.1% formic acid; Buffer B: 0.1% formic acid, 99.9% acetonitrile). The effluent was electrosprayed into the mass spectrometer (LTQ Orbitrap XL, Thermo Fisher Scientific). We analyzed MS/MS data collected from non-deuterated samples to identify the likely sequence of the patent peptides using SEQUEST (Bioworks v3.3.1, Thermo Fisher Scientific) with a peptide tolerance of 8 ppm and a fragment tolerance of 0.1 AMU. We discarded peptides that failed to meet a specified quality score (SEQUEST $P_{pep}$ score <0.99). The peptide data are included in a supplementary table (see *Supplementary file 1*).

A MATLAB-based program, ExMS2, was used to prepare the pool of peptides based on SEQUEST output files. HDExaminer software was next used to process and analyze the HXMS data. HDExaminer identifies the peptide envelope centroid values for non-deuterated as well as deuterated peptides and uses the information to calculate the level of peptide deuteration for each peptide at each timepoint. Each individual deuterated peptide is corrected for loss of deuterium label during HXMS data collection by normalizing to the maximal deuteration level of that peptide, which we measure in a

'full deuterated' (FD) reference sample. The FD sample was prepared in 75% deuterium to mimic the exchange experiment, but under acidic denaturing conditions (0.88% formic acid), and incubated for over 24 hr to allow each amide proton position along the entire polypeptide to undergo full exchange. 20 µL of this reaction was quenched with 30 µL ice-cold FD quench buffer (1 M guanidine hydrochloride, 8% glycerol, and 0.74% formic acid, for a final pH of 2.4–2.6) and rapidly frozen in liquid nitrogen. HDExaminer performs such correction automatically when provided with the FD file. For each peptide, we compare the extent of deuteration as measured in both the on-exchange and FD samples to the maximal number of exchangeable deuterons (maxD) when corrected with an average back exchange level; the median extent of back exchange in our datasets is 18% (*Figure 2—figure supplement 2*).

## HXMS plotting

Peptide plotting was performed in MATLAB, RStudio, and Prism using deuteration levels for each peptide extracted from the HDExaminer outputs. Differences in deuteration levels between two samples were calculated for all peptides for which the identical peptide was found in both conditions, the ND and FD samples. For comparing two different HXMS datasets, we plot the percent difference of each peptide, which is calculated by subtracting the percent deuteration of one sample from that or another, and plotted according to the color legend in stepwise increments (as in *Figure 1F* and *Figure 2—figure supplement 3A*). We include in our figures peptides of identical sequence but different charge states. Although not unique peptides, they do add confidence to our peptide identification as their deuteration levels are in close agreement with each other. Only peptides of high quality and with clear spectra in both HX time courses and 'FD' conditions were included in the analysis. Consensus behavior at each residue was calculated as the average of the differences in HX protection of all peptides spanning that residue (as in *Figure 1F* and *Figure 2—figure supplement 3A*). For the plot of peptide data expressed as the number of deuterons (as in *Figure 2C–D and G*, and *Figure 2—figure supplement 3C–E*), the values are expressed as the mean of three independent measurements±s.d.

## Turbidity assay

Following liquid-liquid phase separation, WT and ISB mutant protein were incubated at room temperature for 100 s prior to UV-visible measurements. Control measurements included protein purification buffer, low salt buffer (50 mM Tris, 75 mM NaCl, 5% glycerol), and WT ISB protein at high salt (25 µM WT ISB, 500 mM NaCl). The optical intensity (turbidity) was measured using a NanoDrop 2000 UV-Vis spectrophotometer (Thermo Fisher Scientific) at 330 nm. The number of replicates is indicated in figure legends.

## Measuring saturation concentration

Following liquid-liquid phase separation, indicated proteins were incubated at room temperature for 100 s and centrifuged at 16,100 × *g* for 10 min to separate the soluble phase from the droplet phase. Then, the entirety of the top phase was removed; the remaining sedimented pellet was resuspended in an equivalent volume of protein purification buffer. 5 µL of both top phase and sedimented pellet, along with a sample of protein after thawing and a sample of protein before sedimentation, were analyzed using SDS-PAGE (4–20% Tris-HCl gradient gel) to determine the saturation concentration. The serial dilution of wild-type ISB, ranging between 0 and 30 µM, was loaded onto a similar SDS-PAGE gel to create a standard curve (with a coefficient of determination $R^2$=0.9), which was used to determine the saturation concentration of ISB. The SDS-PAGE gel was stained with Coomassie Blue and subjected to densitometry using GelQuant Express Analysis Software.

## optoDroplet assay

The plasmid expressing Borealin$_{WT}$-mCherry-Cry2 (*Trivedi et al., 2019*) and a derivative harboring the K→E substitutions at Borealin a.a. 26, 37, and 63 were transfected into HeLa TREx cells that were seeded in 35 mm glass-bottom dishes (Cellvis, D29-20-1.5P). HeLa T-Rex were acquired directly from the manufacturer (Thermo Fisher #R714087), were authenticated by ATCC using STR profiling, and were regularly assessed by the DNA stain uptake method for mycoplasma contamination (and found to be uncontaminated). Lipofectamine 3000 (L3000-008) was used for transfections. Twenty-four hours following transfection, the cells were imaged using a Zeiss Observer-Z1 microscope in the

presence of 5% $CO_2$ in a humidified chamber at 37°C. Cells with similar mCherry expression levels were selected for measurement (phase separation propensity of wild-type and mutant Borealin). To induce phase separation, the cells in the field were exposed to 488 nm light (10 cycles of 100 ms each, with an interval of 30 s between consecutive cycles). The single z-plane mCherry images were acquired immediately after exposure with 488 nm light for 1500 ms during each cycle. The mCherry intensities were measured 270 s after light exposure and quantified in Fiji (ImageJ) software and plotted using the GraphPad Prism software.

## Immunofluorescence

HeLa TREx cells were seeded on coverslips in six-well culture plates. The following day, plasmids (pCDNA5 containing Borealin[WT]-mCherry-Cry2 or mCherry-Cry2) were transfected into HeLa TREx cells as in the optoDroplet assay (see above). After 6 hr, RO-3306 (9 μM) was added to medium to synchronize the cells in late G2-phase. Twenty-four hours following the transfection, the cells were either immediately fixed or exposed to white light on a transilluminator for 10 min to induce liquid-liquid phase separation and then fixed using 4% paraformaldehyde in PHEM buffer (25 mM HEPES, 60 mM PIPES, 10 MM EGTA, and 4 mM $MgCl_2$, pH 6.9) with 0.5% Triton X-100 for 20 min at room temperature in the dark. Following fixation, the coverslips containing the fixed cells were washed three times with PBS. Subsequently, cells were incubated for 30 min at room temperature with blocking buffer (3% BSA, 0.1% Triton X-100 in PBS). Next, the cells were incubated with the following primary antibodies: anti-mCherry pAb (PA5-34974, Invitrogen) and mouse anti-AIM1 mAb (cat 611082, BD Transduction laboratories) in blocking buffer overnight at 4°C. Then, the excess primary antibodies were washed three times using blocking buffer and subsequently incubated with Alexa Fluor 488- and 568-conjugated secondary antibodies (Thermo Fisher) for 90 min. Finally, cells were washed four times with blocking buffer, and the last wash contained DAPI (0.5 μg/mL) in PBS. Cells were mounted with ProLong Gold Antifade and imaged on a Nikon Ti2-E Eclipse Confocal microscope.

## Acknowledgements

This work was supported by NIH grants GM130302 (BEB) and GM134591 (NWB). We acknowledge support of NWB by the UPenn Structural Biology and Molecular Biophysics Training Grant (GM008275). J Shorter (UPenn) provided guidance on phase separation experiments.

## Additional information

### Funding

| Funder | Grant reference number | Author |
| --- | --- | --- |
| National Institute of General Medical Sciences | GM130302 | Ben E Black |
| National Institute of General Medical Sciences | GM134591 | Nikaela W Bryan |
| University of Pennsylvania | Structural Biology and Molecular Biophysics Training Grant GM008275 | Nikaela W Bryan |

The funders had no role in study design, data collection and interpretation, or the decision to submit the work for publication.

### Author contributions

Nikaela W Bryan, Conceptualization, Formal analysis, Investigation, Writing – original draft, Writing – review and editing; Aamir Ali, Leland Mayne, Investigation, Writing – review and editing; Ewa Niedzialkowska, Formal analysis, Writing – review and editing; P Todd Stukenberg, Conceptualization, Writing – review and editing; Ben E Black, Conceptualization, Supervision, Funding acquisition, Writing – original draft, Project administration, Writing – review and editing

## Author ORCIDs

Nikaela W Bryan (iD) https://orcid.org/0000-0002-5293-5145
Leland Mayne (iD) http://orcid.org/0000-0001-6969-0474
P Todd Stukenberg (iD) http://orcid.org/0000-0002-6788-2111
Ben E Black (iD) http://orcid.org/0000-0002-3707-9483

## Decision letter and Author response

Decision letter https://doi.org/10.7554/eLife.92709.sa1
Author response https://doi.org/10.7554/eLife.92709.sa2

---

# Additional files

## Supplementary files

• Supplementary file 1. Hydrogen-deuterium exchange mass spectrometry summary table for ISB free protein and ISB droplet protein datasets.

• Supplementary file 2. List of gene block sequences used to produce $I_{Mut1}SB$, $I_{Mut2}SB$, $I_{Mut3}SB$, $I_{Mut4}SB$, $I_{Mut5}SB$, $ISB_{Mut}$, and $I_{Mut6}SB_{Mut}$ mutant proteins.

• MDAR checklist

## Data availability

Source data are provided with this paper. The HXMS data in this study has been deposited in the Pride database under accession code PXD034374. The structure 2QFA (https://doi.org/10.2210/pdb2QFA/pdb) from the Protein Data Bank (https://www.rcsb.org/) was used in this study. An AlphaFold prediction for the Borealin protein (primary accession number Q53HL2) was used in this study.

The following dataset was generated:

| Author(s) | Year | Dataset title | Dataset URL | Database and Identifier |
| --- | --- | --- | --- | --- |
| Bryan N, Black BE | 2023 | HXMS data | https://www.ebi.ac.uk/pride/archive/projects/PXD034374 | PRIDE, PXD034374 |

The following previously published dataset was used:

| Author(s) | Year | Dataset title | Dataset URL | Database and Identifier |
| --- | --- | --- | --- | --- |
| Jeyaprakash AA, Klein UR, Lindner D, Ebert J, Nigg EA, Conti E | 2007 | Crystal structure of a Survivin-Borealin-INCENP core complex | https://www.rcsb.org/structure/2qfa | RCSB Protein Data Bank, 2QFA |

---

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
