## [Editor Report]

This study is important for the phase separation field as it demonstrates that hydrogen/deuterium-exchange mass spectrometry (HXMS) can identify key regions important in driving liquid-liquid demixing. The authors convincingly confirm their HXMS results by mutagenesis. The study uses the chromosomal passenger complex (CPC) as an example, but the methodology will be applicable to other proteins or protein complexes undergoing liquid-liquid demixing.

---

## [Decision Letter]

[Editors' note: this paper was reviewed by Review Commons.]

Thank you for submitting your article "Structural Basis for the Phase Separation of the Chromosome Passenger Complex" for consideration by *eLife*. Your article has been reviewed by 2 peer reviewers at Review Commons, and the evaluation at *eLife* has been overseen by a Reviewing Editor and Detlef Weigel as the Senior Editor, in consultation with the original reviewers from Review Commons.

Your manuscript shows (with the caveats below, which we hope can be addressed) that hydrogen/deuterium-exchange followed by mass spec (HXMS) is a useful tool to understand relevant interactions for liquid-liquid demixing, at least when folded domains are involved.

Based on the previous reviews and the revisions, the manuscript has been improved but there are two remaining issues that we think need to be addressed:

(1) The characterization of the mutants using DIC and absorbance at a single time point (Figure 3C/D, 4C/D, etc.) seems insufficient-especially since Figure 1E shows that droplets can become fewer at later timepoints. In S1B you show a phase diagram for the wild-type construct. In our opinion, an analogous phase diagram for key mutants (such as Imut6SB, ISBmut, and Imut6SBmut) is required to make the conclusions from the mutant analysis solid.

(2) In the optoDroplet assay that you use as in vivo evidence, it is critical to demonstrate that INCENP and Survivin are present in these droplets and/or that the Borealin constructs that have been introduced from a 1:1:1 complex with INCENP and Survivin, without an excess of free Borealin. Since the relevant residues identified are at the interface between trimers, the in vivo observations are only relevant if there are indeed trimers present.

The fact that it is contested whether the CPC forms functionally relevant biomolecular condensates in vivo does in our opinion not take away from the main message of your manuscript that HXMS could be an additional useful technique to understand liquid-liquid demixing.

[Editors' note: further revisions were suggested prior to acceptance, as described below.]

Thank you for resubmitting your work entitled "Structural Basis for the Phase Separation of the Chromosome Passenger Complex" for further consideration by *eLife*. Your revised article has been evaluated by Detlef Weigel (Senior Editor) and a Reviewing Editor.

The manuscript has been improved but minor remaining issues need to be addressed, as outlined below:

(1) An image screen by *eLife* staff found a partial overlap between the WT images in Figure 1B and 3C. This is not a problem, since the same data can be used for both and the source image is identical. But to avoid confusion, we suggest that your legend for Figure 3 mentions that the WT experiment is the same as in Figure 1. And maybe it would be better to show the identical region of the source image, rather than two different, partly overlapping regions. The choice if yours.

(2) Evidence for incorporation of the mutant Borealin-mCh-Cry2 into the CPC is still missing. Do the weaker foci shown in Figure 6B co-localize with Aurora B, or does mutant Borealin-mCh-Cry2 localize to centromeres in mitosis? This could also be done by an immunoprecipitation if the imaging is uninformative.

(3) For the interpretability of the cellular experiment, it is critical that the levels of WT and mutant Borealin are similar. The authors mention that cells with similar mCherry expression levels were selected. Could there be a control for this, such as total cellular mCherry levels in wild-type and mutant? Or could the authors confirm that the images in Figure 6B are taken with the same imaging conditions and scaled the same for display? (Adding this information to the figure legend or methods section would be helpful.)

Related, but less important: Could no light/light for mutant Borealin-mCh-Cry2 be shown in Figure 6 —figure supplement 1 for completeness?

Typos that could be fixed while revising:

- Line 97 "mechanisms …is" should be "mechanisms… are"

- Line 421 "boreain"

- Line 421: Please spell out full length (FL).

- Line 439 remove dash before WTISB_R.

- Line 494: Please spell out ND – presumably non-deuterated.

- Line 570: "cells were quantified for the time of appearance of mCherry foci in the nucleus." The intensity of foci is quantified (Figure 6C), but not their time of appearance – so wording should probably be changed. Additionally mentioning when after light exposure the intensity was measured would be useful.

- Line 724 "potted"

- Line 782 "I well"

- Line 807: presumably "Ile" rather than "Ilu"?

---

## [Author Response]

Referee #1:The authors have revised the manuscript satisfactorily. The work presented in the manuscript convincingly establishes HXMS as a useful tool to characterise molecular interactions driving liquid-liquid demixing.

As the reviewer kindly stated in the initial review, our paper represents a ‘…highly relevant and significant work, particularly with the rapidly growing list of examples for Phase separation of protein/protein assemblies and their potential biological roles’ and that our experiments ‘solidly and convincing establish HXMS as a useful tool to characterize molecular interactions driving liquid-liquid demixing’ and ‘Considering its applicability to characterize wide-ranging protein assemblies implicated in phase separation, this work will be of interest to a broad readership’. We appreciate that the Reviewer is now satisfied with our revisions.

Referee #2:In the revised version of the manuscript, the authors have now incorporated the necessary minor changes. The authors have also added interesting in vivo results correlating the presence of salt bridges in the ISB complex to its phase separation propensity inside cells.

We thank the reviewer for recognizing that we have made the necessary minor changes, for recognizing the interesting nature of our in vivo results, and for the additional comments.

'By the latest timepoint, 3000 s, there was some diminution in the number of droplets (Figure 1E), which may indicate the start of a transition of the droplets to a more solid state (i.e., gellike).' The authors suggest that this transition from a liquid to gel-like state is a reasonable conclusion here and extends beyond the scope of this study. It will be an interesting direction to pursue to see the evolution of the material property of these condensates as a function of time to ascertain that this transition happens and if that is somehow relevant to its localization and subsequent function at the centromere.

We thank the reviewer for the suggestion for further investigation to follow on the findings we report.

It might still be pertinent to just include a comment as to why the condensates formed from the ISB complex diminish in size at 1000s (Figure 1E) and whether any of the parameters such as, circularity, droplet size, droplet intensity, as mentioned by the authors in Response no. 9, affect these measurements, irrespective of sedimentation assays showing similar concentration distributions between the condensed and dispersed phases, as it was clarified by the authors that sedimentation was not performed prior to HXMS measurements.

We have now modified the relevant part of the Results section, as follows:

Page 9, 1st Paragraph in the main manuscript, ISB heterotetramers should be replaced with ISB heterotrimer.

This is now corrected.

Just a clarification, is the BorealinWT and BorealinMut-mCherry-Cry2 construct transfected into cells, the full length Borealin (280 aa) harboring the respective mutations, or is it the exact same construct used in the in vitro biochemical studies? Should be explicitly mentioned.

This is now added to explicitly mention.

The addition of the in vivo cellular data where disruption of salt-bridges reduces phase separation in the cellular environment, is a good addition in the current version of manuscript and I am sure that is going to be a good starting point for future studies.

We thank the reviewer for the encouragement moving forward.

Overall, this paper brings forward a useful technique to probe the conformational landscape of proteins in the condensed droplet phase and compare it with its dispersed phase. This paper serves as an interesting read showing how specific salt-bridge interactions between multiple stoichiometric protein complexes can be the driving force for phase separation and definitely forms the basis for future studies in a more physiologically relevant context

We thank the reviewer once again for the positive assessment of our work.

[Editors’ note: what follows is the authors’ response to the second round of review.]

Your manuscript shows (with the caveats below, which we hope can be addressed) that hydrogen/deuterium-exchange followed by mass spec (HXMS) is a useful tool to understand relevant interactions for liquid-liquid demixing, at least when folded domains are involved.Based on the previous reviews and the revisions, the manuscript has been improved but there are two remaining issues that we think need to be addressed:(1) The characterization of the mutants using DIC and absorbance at a single time point (Figure 3C/D, 4C/D, etc.) seems insufficient-especially since Figure 1E shows that droplets can become fewer at later timepoints. In S1B you show a phase diagram for the wild-type construct. In our opinion, an analogous phase diagram for key mutants (such as Imut6SB, ISBmut, and Imut6SBmut) is required to make the conclusions from the mutant analysis solid.

Each of the specific mutants mentioned, and other relevant ones, underwent the DIC and absorbance experiments noted, as well as a saturation concentration experiment that involves sedimentation. The practical aspects of the experiments means that we assay across a time span that goes from short time points (absorbance taken at about 100s after assembling the reaction) to intermediate ones (DIC at ~5 minutes after assembling reaction including the time to get onto the microscope) to sedimentation (where samples are isolated a total of ~12 minutes after initial assembling the reaction including the time for the actual sedimentation). Thus, if the mutants were altogether changed by changes in droplet behavior, it would have to have happened out past 12 minutes (and certainly past where any phase diagram experiment would be performed) and spanning the majority of the timepoints that we used to initially map where the relevant structural features of the ISB_WT_ were protected by droplet formation. We thank the editors for suggesting that the present presentation passed over this important aspect, and in response we have highlighted the 3 distinct ways in which the mutants were assessed and how this spans timepoints past when the HX changes were measured.

Added on pg 9 of the Results section:

“We emphasize that the key mutants that break (including I_mut6_SB and ISB_mut_) and re-form (I_mut6_SB_mut_) the salt-bridge-mediated droplet formation underwent three different measurements (turbidity [Figures 4 and 5], microscope-based detection [Figures 4 and 5], and sedimentation-based determination of saturation concentration [Figures 5 and Figure 5 —figure supplement 1]). These different types of measurements vary by several minutes due to practical considerations relative to the time from initial reaction assembly. Thus, our conclusions about the phase separation of mutant versions of ISB heterotrimers are based on independent experiments that span the time window covered in our initial HXMS analysis of ISB_WT_ (Figures 1 and 2).”

(2) In the optoDroplet assay that you use as in vivo evidence, it is critical to demonstrate that INCENP and Survivin are present in these droplets and/or that the Borealin constructs that have been introduced from a 1:1:1 complex with INCENP and Survivin, without an excess of free Borealin. Since the relevant residues identified are at the interface between trimers, the in vivo observations are only relevant if there are indeed trimers present.

We provide evidence that strongly indicates that the entire CPC is engaged with the

Borealin-mCherry-Cry2 fusion protein, forming droplets in the nucleus. See the all-new Figure 6A, Figure 6 —figure supplement 1, and Figure 6 —figure supplement 2. We detected the CPC by assessing the localization of endogenous Aurora B, which is well understood to be directly bound to INCENP, and then through INCENP, indirectly to Survivin and Borealin. With the stronger focus now justified on inducible nuclear foci, we restricted our conclusions to that compartment, thereby re-analyzing the WT vs Mut data, as shown in the revised panel Figure 6C.

The fact that it is contested whether the CPC forms functionally relevant biomolecular condensates in vivo does in our opinion not take away from the main message of your manuscript that HXMS could be an additional useful technique to understand liquid-liquid demixing.

Thank you for your work to guide us to extend and improve the manuscript in the manner described, above.

[Editors’ note: what follows is the authors’ response to the third round of review.]

The manuscript has been improved but minor remaining issues need to be addressed, as outlined below:(1) An image screen by eLife staff found a partial overlap between the WT images in Figure 1B and 3C. This is not a problem, since the same data can be used for both and the source image is identical. But to avoid confusion, we suggest that your legend for Figure 3 mentions that the WT experiment is the same as in Figure 1. And maybe it would be better to show the identical region of the source image, rather than two different, partly overlapping regions. The choice if yours.

We thank the *eLife* staff for identifying this unintentional partial overlap, and we appreciate the thoughts on how to resolve the issue. We have other independent images from the same experiment, so we chose a completely different one now for the WT data shown in Figure 3C. That panel is new, along with the requested mention added to the legend, and the corresponding source data file is updated to match.

(2) Evidence for incorporation of the mutant Borealin-mCh-Cry2 into the CPC is still missing. Do the weaker foci shown in Figure 6B co-localize with Aurora B, or does mutant Borealin-mCh-Cry2 localize to centromeres in mitosis? This could also be done by an immunoprecipitation if the imaging is uninformative.

We have added a new panel to Figure 6 —figure supplement 1 that clearly shows that Aurora B colocalizes with mutant Borealin-mCh-Cry2. We originally left this out because the difference in intensity between WT and mutant Borealin in the fixed-cell IF is not a distinct as in the live cell experiments in Figure 6B. We feel the difference is due to the way the cells are exposed to light. In the Figure 6, we use our microscope to expose cells to a controlled amount of light for the indicated times. In contrast, for IF as in Figure 6 —figure supplement 1A, we need to expose the whole coverslip so we simply put the cells on a light box for 10 minutes. This will expose cells to a tremendous amount of light leading to saturation effects. We have modified the figure legend to explain these differences.

(3) For the interpretability of the cellular experiment, it is critical that the levels of WT and mutant Borealin are similar. The authors mention that cells with similar mCherry expression levels were selected. Could there be a control for this, such as total cellular mCherry levels in wild-type and mutant? Or could the authors confirm that the images in Figure 6B are taken with the same imaging conditions and scaled the same for display? (Adding this information to the figure legend or methods section would be helpful.)Related, but less important: Could no light/light for mutant Borealin-mCh-Cry2 be shown in Figure 6 —figure supplement 1 for completeness?

While performing the experiment, we chose cells that had similar mCherry intensities. In the images that we chose, we also show cells taken with identical imaging conditions and displayed with identical settings (the figure legend now states this as well). To directly address your concern we include a new panel (B) in Figure 6 —figure supplement 1, where we measured the nuclear mCherry intensity before the light exposure of each of the cells used to quantify the foci intensity in Figure 6C. The differences in the initial mCherry staining are not significant (p=0.9) between the WT and MUT constructs, but there is a significant difference in the intensity of foci formed by light.

Typos that could be fixed while revising:- Line 97 "mechanisms …is" should be "mechanisms… are"- Line 421 "boreain"- Line 421: Please spell out full length (FL).- Line 439 remove dash before WTISB_R.- Line 494: Please spell out ND – presumably non-deuterated.- Line 570: "cells were quantified for the time of appearance of mCherry foci in the nucleus." The intensity of foci is quantified (Figure 6C), but not their time of appearance – so wording should probably be changed. Additionally mentioning when after light exposure the intensity was measured would be useful.- Line 724 "potted"- Line 782 "I well"- Line 807: presumably "Ile" rather than "Ilu"?

All typos have been fixed and the additional mention of the timing of intensity measurements was added as requested.